# Laughlin anyon complexes with Bose properties

L. V. Kulik[1,2], A. S. Zhuravlev[1], L. I. Musina [3,4 ✉], E. I. Belozerov[1,2], A. B. Van'kov[1,2], O. V. Volkov[1], A. A. Zagitova[1], I. V. Kukushkin[1] & V. Y. Umansky[5]

Two-dimensional electron systems in a quantizing magnetic field are regarded as of exceptional interest, considering the possible role of anyons—quasiparticles with non-boson and non-fermion statistics—in applied physics. To this day, essentially none but the fractional states of the quantum Hall effect (FQHE) have been experimentally realized as a system with anyonic statistics. In determining the thermodynamic properties of anyon matter, it is crucial to gain insight into the physics of its neutral excitations. We form a macroscopic quasi-equilibrium ensemble of neutral excitations - spin one anyon complexes in the Laughlin state $\nu = 1/3$, experimentally, where $\nu$ is the electron filling factor. The ensemble is found to have such a long lifetime that it can be considered the new state of anyon matter. The properties of this state are investigated by optical techniques to reveal its Bose properties.

[1] Institute of Solid State Physics Russian Academy of Sciences Chernogolovka, Moscow District, 2 Academician Ossipyan Street 142432, Russia. [2] National Research University Higher School of Economics, Moscow, 20 Myasnitskaya Street 101000, Russia. [3] Moscow Institute of Physics and Technology, Dolgoprudny 141701, Russia. [4] Skolkovo Institute of Science and Technology, Bolshoy Boulevard 30, bld. 1 Moscow 121205, Russia. [5] Braun Center for Submicron Research, Weizmann Institute of Science, 234 Herzl Street, POB 26 Rehovot 76100, Israel. ✉email: musina.li@phystech.edu

Seminal work[1] on the experimental discovery of quasi-particles that obey anyonic statistics in the electron Laughlin state $\nu = 1/3$ [2] opened fundamentally new prospects for studying the objects, the very existence of which has been discussed thus far merely hypothetically. Surprisingly, the anyonic statistics of FQHE quasiparticles were predicted immediately following Laughlin's pioneering works[3,4]. However, it took over thirty years before there was reported the first direct experimental evidence in support of this remarkable physical phenomenon, whose practical application can offer entirely new perspectives in solid-state physics. As for the quasiparticle systems that obey anyonic statistics, there have been predicted a number of intriguing results, unachievable in boson and fermion systems[5,6], while considering the non-abelian anyons in the FQHE state of 5/2 has already led to a beautiful theoretical idea of topological quantum computation[7–9].

Although experimental information about the charged quasiparticles can be obtained using magneto-transport techniques, our understanding of neutral excitations in FQHE states, inactive in transport experiments, is relatively limited. Brilliant theoretical predictions about one type of neutral excitations, magneto-rotons in a single-mode approximation[10,11], have been confirmed by microwave absorption experiments, both qualitatively and quantitatively, for several FQHE states with broken translation symmetry[12]. However, it is impossible to draw any conclusions about the statistical properties of magneto-rotons based on these experiments, as such excitations have a short lifetime[12]. Moreover, it is rather challenging to understand from the reported experiments[12,13] whether magneto-rotons constitute the major excitation branch or there are other yet unknown excitations that might be of great importance to the statistics and thermodynamics of FQHE states. The existing theories hypothesize about various properties of neutral excitations and even draw parallels between the magneto-roton complexes in FQHE (magneto-gravitons) and the gravitons in the general theory of relativity[14–18]. Most daring theories suggest that the physics of FQHE excitations helps broaden our outlook on the evolution of the Universe[19,20].

The recipe for constructing quasi-equilibrium ensembles of magneto-rotons in two-dimensional electron systems under stationary excitation by the photons of a specific energy range, has already been developed for the integer Hall insulators[21]. It turns out that the ensembles can be formed provided that the physical characteristics of the electron system meet certain requirements. For example, the lowest-energy excitations should change the total spin of the electron system. In that case, the quasi-equilibrium ensembles of neutral excitations having a roton minimum in their dispersion are formed as a result of drastic retardation in the spin-flip process in the magnetic field, with the simultaneous conservation of the large roton momentum[21]. Seemingly, this situation is unlikely to occur in the Laughlin state $\nu = 1/3$ presumably analogous to the integer Hall ferromagnet $\nu = 1$, though in terms of the composite fermions rather than electrons[22] (for the Hall ferromagnet $\nu = 1$, the aforementioned requirements are definitely not satisfied due to absence of the roton minimum in the dispersion of the spin exciton[23]). The single-mode approximation proven successful in describing the spin-zero neutral excitations in the Laughlin state $\nu = 1/3$ also predicts a monotonous dispersion dependency for spin one neutral excitations at $\nu = 1/3$, similar to that of a spin exciton in the Hall ferromagnet $\nu = 1$ [24].

Nevertheless, finding the exact solution to Schrodinger's equation for a multiparticle electron system[25] gave the first indication that in the Laughlin state $\nu = 1/3$, the spin excitations with momentum on the order of inverse magnetic length can have lower energy than the minimal energy of the magneto-rotons. This result did not account for a substantial contribution to the energy of the excitations from the single-particle Zeeman energy that is linearly dependent on the magnetic field. Therefore, so far, it has not been clear whether physical systems with the spin one neutral excitations can, in practice, have lower energy than magneto-rotons.

In the presented work, we develop a numerical scheme of solving Schrodinger's equation for the electron system with a finite number of particles in line with previous studies[26] and investigate the dispersion properties of neutral excitations for the Laughlin state $\nu = 1/3$ in GaAs/AlGaAs quantum wells. The single-mode approximation is found inapplicable for describing the spin one branches of the excitations in this case. Moreover, we confirm that for specific parameters of the electron system, the spin one neutral excitations are indeed the lowest-energy excitation branches (see "Methods" section). Given the selected electron system with necessary parameters, by means of optical techniques (photoluminescence and photoinduced resonant reflection) (Fig. 1), we succeeded in forming a macroscopic ensemble of neutral spin one excitations in the Laughlin state $\nu = 1/3$ and devised a method for the direct measurement of their relaxation time. Our findings show that the ensemble of spin one neutral excitations exhibits the properties of a Bose system.

## Results and discussion

Let us look at the photoluminescence first as namely this technique is used to form a quasi-equilibrium ensemble of neutral excitations (see "Methods" section). The photoluminescence signal at the formation of the Laughlin state $\nu = 1/3$ undergoes significant changes. At low temperatures, the photoluminescence intensity from the upper spin sublevel is an order of magnitude higher than that from the lowest spin sublevel filled with electrons in the equilibrium (Fig. 2). Moreover, at higher temperatures (above 1 K), when both spin sublevels are filled in the equilibrium and the Laughlin state $\nu = 1/3$ is destroyed, the photoluminescence intensity from the lowest spin Landau sublevel exceeds that from the upper spin sublevel, which corresponds to the probabilities of optical transitions for analogous electronic systems[27]. The effect of enormous gain in the photoluminescence signal in the Laughlin state $\nu = 1/3$ produced by the excited spin Landau sublevel has universal nature, i.e., it is independent of the photon energy exciting the electron system (Fig. 3). Such an amplification of the photoluminescence signal generated by the excited spin Landau sublevel is nothing else but the collective response of the quasi-equilibrium excitations to a valence hole, analogous to that already observed in the photoluminescence signal of another condensate (the condensate of spin-triplet magneto-rotons demonstrating properties of a Bose system) in the integer quantum Hall insulator $\nu = 2$ [21]. Based on measured photoluminescence spectra, we can draw a qualitative conclusion that under continuous wave (cw) photoexcitation, there is formed a dense ensemble of neutral excitations, exhibiting collective response to a photoexcited hole in the valence band. Besides, all experimentally observed optical transitions from the zero Landau level of heavy holes of the valence band occur at the upper spin sublevel of the zero Landau level in the conductance band (see the experiment and the theory in Supplementary Materials). Therefore excitations with the change of the spin of the whole electron system can be created in our experiments only. Furthermore, analyzing lifetimes for different spin one excitations[28], it can be concluded that spin one "magneto-gravitons" are the most likely candidates to form this ensemble (see "Methods" section).

Our further study is devoted to determining the statistical properties of the thus formed ensemble of excitations. To accomplish this, we employed the technique of photoinduced

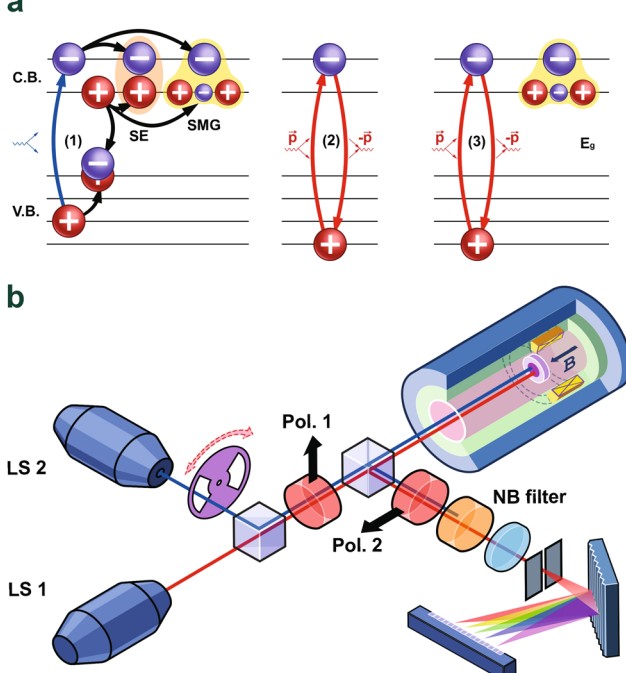

**a**

**b**

**Fig. 1 Schematic diagram of the optical setup. a** The optical processes utilized in the experiment. Left (1): Photoluminescence (PL) with the formation of neutral excitations. An electron from the valence band is excited into the higher spin sublevel of the zero Landau level (0LL) of the conductance band; then the resulting hole relaxes into the state of minimal energy—0LL of heavy holes (hh); an equilibrium electron from the lower spin sublevel of 0LL of the conductance band recombines with this hole; The excited electron on higher spin sublevel and a hole left at the lower spin sublevel after the recombination form a spin exciton (SE). If during the recombination process a part of the energy is redistributed into an excitation in the lower spin state the resulting excitation is spin-magneto-graviton (SMG). Middle (2): Resonant reflection from the unexcited system. From 0LL of hh, an electron is excited into the higher spin sublevel of the conductance band and immediately (phase conserving process) returns back to the valence band. Right (3): Photoinduced resonant reflection. The process is similar to (2), but the electron is transferred in presence of a spin-zero excitation in the ground state. **b** The scheme of the experimental setup is shown. Pump (LS 2) and probe (LS 1) signals go through the optical scheme into the cryostat and excite the sample. The reflected signal is decomposed into a spectrum on a diffraction grating. For the extended description of the experimental setup and the processes, see in "Methods" section and Supplementary Materials.

resonant reflection—the photoinduced elastic scattering of light. In essence, it means using one radiation source to form an ensemble of excitations, whereas the other provides the probing photons that are elastically scattered on the neutral excitations (Fig. 1). By varying the excitation wavelength of the former at a fixed wavelength and the radiation power level of the latter, we can compare the effectiveness of forming neutral excitations by the photons of different energies. Conversely, varying the wavelength of the probing radiation from the second radiation source while keeping constant the wavelength and the power level of the first one makes it possible to investigate how effective the different channels of elastic light scattering are, based on the photon scattering efficiency from the fixed quantity of the excitations created by the first radiation source.

In the presence of a macroscopic ensemble of excitations, a standard signal of photoinduced resonant reflection has a negative sign (Fig. 4), demonstrating the filling of the phase space on the upper spin Landau sublevel at the appearance of the electrons incorporated in the excitations. This, in turn, makes the electrons less likely to transfer from the valence band to the upper spin sublevel. Accordingly, the depth of the minimum in the scattering signal dependency on the excitation photon energy characterizes the effectiveness of pumping the excitations with photons from the first radiation source as a function of the photon energy.

Apart from the standard channel of elastic light scattering, the spectrum of resonant reflection evidences the development of another scattering channel. The scattering efficiency in this channel increases with the rising number of neutral excitations. Its energy exceeds that of the major scattering channel, forming an electron–hole pair, consisting of a valence hole and an electron at the upper spin sublevel of the zero Landau level of the conductance band, by the calculated energy of a spin one "magneto-graviton" with zero momentum (1.4 meV) (Fig. 5). As the quantity of excitations in the electron system increases, the amplitude of the scattering signal in this channel is amplified nearly by order of magnitude. Note that in this case, the radiation power is raised only in the first radiation source used to pump the neutral excitations, whereas that of the probing source remains unchanged (Fig. 5). Therefore, the more excitations are present in the electron system, the more likely photon backscattering is—a particular property of the Bose statistics[29]. One of the plausible scenarios for such a scattering channel can be as follows. A photon from the probing radiation source is absorbed in a quantum well, yielding an electron–hole pair and a spin one "magneto-graviton" with zero momentum (Fig. 1a). After that, the electron–hole pair recombines, absorbing the newly created spin one "magneto-graviton" or another spin one "magneto-graviton" occupying the same quantum state, which leads to a scattering signal with the energy and longitudinal momentum of the probing photon (the transverse component to the quantum well plane is not preserved as the translation symmetry in the direction of the heterostructure growth is broken). Given the Bose statistics of spin one "magneto-gravitons," the probability of this process is proportional to the number of spin one "magneto-gravitons" occupying the same quantum state (scattering probabilities for similar processes are discussed in ref. [16], besides the complex valence band structure may facilitate the scattering efficiency in this channel, see Supplementary Materials). Presumably, there are possible alternative descriptions of the observed scattering process, though all of them would be related to forming the system of neutral anyon complexes with Bose statistics[29].

The most striking outcome of the resonant reflection measurements concerns the relaxation time of the ensemble of neutral excitations. To measure the relaxation time directly, the first laser source is cut off by an external mechanical shutter (trigger time of 10 μs), and the amplitude change in the resonance reflection signal due to the second (probing) source is registered as a function of the time elapsed from the triggering of the shutter. Despite the constant level of probing radiation power, the signal of resonant reflection is reduced with time. The decay time of the resonance reflection signal exceeds 10 s (Fig. 6), which is an absolute record among all the relaxation times of excitations in quantum Hall states known to this day[21]. Such a prolonged relaxation time provides the grounds for considering this ultra-long-life quasi-equilibrium ensemble (a dense ensemble of neutral anyon complexes with spin one) as a new state of anyon matter. An experimental realization of such an ensemble of neutral excitations exhibiting collective response presented in this paper opens up intriguing possibilities of directly manipulating the quasiparticles of this exotic state of matter in real-time.

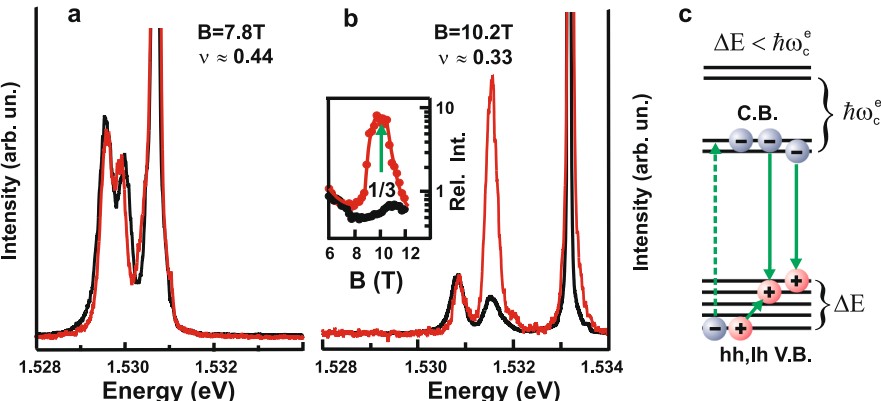

**Fig. 2 Photoluminescence spectra for two values of electron filling factor.** $\nu = 0.44$ (**a**) and 1/3 (**b**), measured at two temperatures of the electron system: 0.5 K, required for the Laughlin state $\nu = 1/3$ (red lines), and 1.6 K, where the Laughlin liquid $\nu = 1/3$ is already destroyed (black lines). The photon excitation energy is 1.534 eV. Photoluminescence intensities are normalized by those of the lowest spin sublevel. The inset in **b** shows the amplitude ratio of photoluminescence signals from the upper and lower spin sublevels as a function of the magnetic field at 0.5 K (red dots) and 1.6 K (black dots). The solid lines are added for convenience. The pictorial diagram **c** illustrates the process of excitation and recombination of the electron system. The electron is excited from a heavy (light) holes energy level (hh, lh) to the conductance band (C.B.) upper spin sublevel. The hole moves to its lowest-energy position and recombines with an equilibrium electron from the lower spin sublevel. The energy of the excitation radiation is chosen to be lower than that of the optical transition from the first electron Landau level to the valence band V.B. (for a more detailed explanation, see Supplementary Materials).

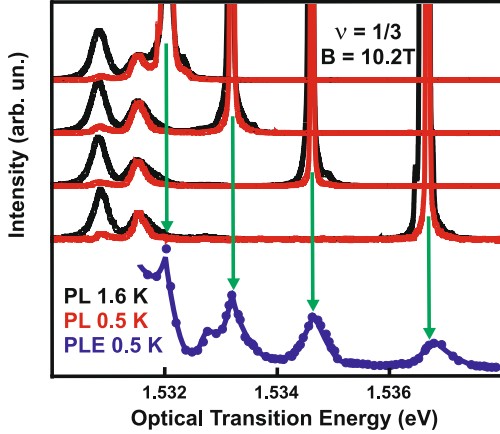

**Fig. 3 Photoluminescence spectra of the Laughlin liquid $\nu = 1/3$ measured for different excitation energies.** Spectra given for the temperatures of 0.5 K (red lines) and 1.6 K (black lines). The energies for exciting the electron system are obtained from the photoluminescence excitation (PLE) spectra (blue dots and the solid line below). Regardless of the excitation resonance, the spectra exhibit no significant variations, indicating the universal nature of the change in the amplitude of photoluminescence (PL) signal observed at the forming of the Laughlin state $\nu = 1/3$. Green arrows indicate the laser energy exciting the PL signal.

## Methods

**Excitations classification.** Magneto-rotons (MR) and magneto-gravitons (MG) are spin-zero excitations known from literature[10]. Since in our experiment only the spin-flip excitations are active (Supplementary Materials), we work with the spin-one analogs of these known excitations: spin-magneto-roton (SMR) and spin-magneto-graviton (SMG). Cartoons are available on Supplementary Figs. 2 and 3.

**Computation.** To demonstrate the accuracy of our computation scheme, we compare the exact analytical calculation of the spin exciton dispersion for the Hall ferromagnet $\nu = 1$[23] to the numerical solution of Schrodinger's equation for a system of 28, 29, and 30 electrons (for calculation details, see Supplementary Materials). The results of numerical calculation closely match the exact analytical solution (Supplementary Fig. 1).

The next step in testing our computation scheme is comparing the solution to Schrodinger's equation for a finite number of particles (7, 8, and 9) in the Laughlin state $\nu = 1/3$ to the dispersion dependence of magneto-rotons obtained in the

single-mode approximation[11]. For the momenta less than inverse magnetic length, the numerical calculations are practically identical to those based on the analytical treatment of the single-mode approximation (Supplementary Fig. 2). Furthermore, we confirm the prediction made in ref. [15] regarding the energy of the magneto-roton branch of excitations falling into the continuum of low-momentum "magneto-gravitons"—the quadrupole oscillations of electron density with the total angular moment of 2[15]. We note that in the following discussion, this type of neutral excitations is referred to as "magneto-gravitons" in consistency with theoretical nomenclature established in the scientific literature on such excitations[15–17]. We also reproduce the minimum in the dispersion dependency of magneto-rotons in the vicinity of one-and-a-half inverse magnetic lengths, though it is deeper than predicted by the single-mode approximation.

The central result of our computation is the appearance of an excitation branch (spin one magneto-roton) with lower energy than that of magneto-rotons. To explore the nature of this branch, we consider the dispersion of three lowest-energy excitations with spin one (Supplementary Fig. 3). It turns out that besides the spin exciton (spin reversal on the zero Landau level) discussed in the single-mode approximation[24], the spectrum contains the combined excitations of the spin and charge oscillations. The spin-charge branches (a spin flip with simultaneously induced charge density oscillations at the lowest spin sublevel of zero Landau level) exhibit negative dispersion, which, at some momentum, crosses over the dispersion of the spin exciton branch.

The physical reason for such behavior of the spin one excitation branches, as well as the difference between the behavioral patterns of spin one and spin-zero excitation branches, are quite understandable. As for the spin-zero branches, the "magneto-gravitons" with near-zero momenta represent the complexes of two magneto-rotons with oppositely directed momenta of equal magnitude[28]. Thus, their energy cannot be lower than the doubled energy of the roton minimum (neglecting the interaction between the magneto-rotons) achieved precisely for zero-momentum "magneto-gravitons". The rest of the combinations of magneto-rotons have larger energy, leading to the positive dispersion of "magneto-gravitons" (Supplementary Fig. 2). Conversely, the spin one "magneto-rotons" should have minimal energy near the roton minimum of spin-zero magneto-rotons. In this case, they are combinations of a spin exciton with zero momentum (and zero Coulomb's energy according to Larmor's theorem) and a spin-zero magneto-roton. Consequently, the dispersion dependency of spin one "magneto-gravitons" (combination of spin zero and spin one magneto-rotons) becomes negative at near-zero momenta. At some momentum, the energies of spin one "magneto-gravitons" and spin exciton intersect, which leads to their anti-crossing and formation of local minima for the intersecting excitation branches. Accordingly, at near-zero momenta, the lowest-energy spin branch behaves as a spin exciton one, although for momenta on the order of inverse magnetic length, it becomes a spin one magneto-roton branch, and, due to the anti-crossing repulsion, its energy ends up being the lowest among the rest of the neutral excitation energies.

Last but not least, we check whether the spin excitation branches will still have the lowest energy if we take into consideration the actual physical characteristics of the quantum well hosting the electron system—the geometric form factor weakening Coulomb interaction[30] and the linear contribution of the single-particle Zeeman energy[31]. We find that it is possible for the spin-zero magneto-roton to be the lowest-energy excitation in the region of roton minimum. On the other hand, there is also a possibility that the energy of the spin one excitation branch can be

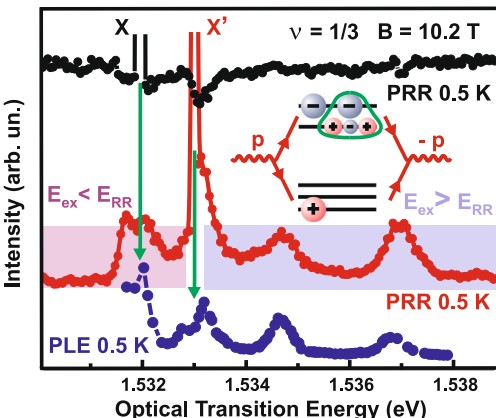

**Fig. 4 Spectra of photoinduced resonant reflection.** The spectra of PRR are obtained by varying the wavelength of the first radiation source used to create excitations, given a fixed wavelength and power level of the probing (second) source. Black and red dots indicate, respectively, a standard channel of elastic light scattering and the nonstandard one where the amplitude is growing with the formation of neutral excitations. In the latter case, the enhanced response is observed at different photon energies of the first source. The inset illustrates a possible process of elastic light scattering in the nonstandard channel. The resonances for exciting the electron system are obtained from the photoexcitation spectra (blue dots and the solid line below, PLE photoluminescence excitations). X and X' are laser positions of the probing laser for the standard and nonstandard elastic light scattering channels (pointed by the green arrows for convenience). Laser power is 10 μW at the excitation spot size of 100 μm. $E_{ex}$ and $E_{rr}$ stand for excitation energy and resonant reflection energy, respectively.

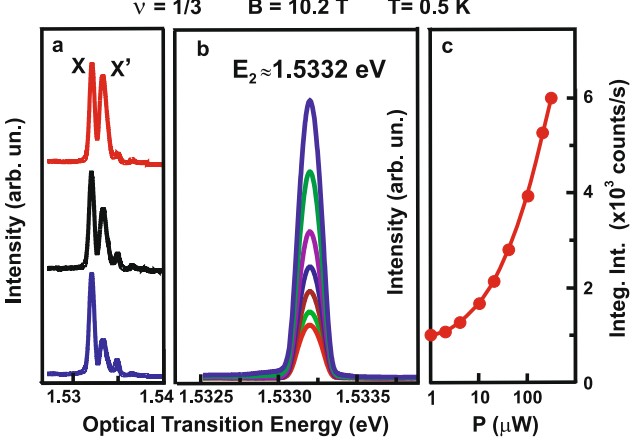

**Fig. 5 Resonant reflection. a** Spectra of photoinduced resonant reflection obtained by varying the wavelength of the probing (second) radiation source at a fixed wavelength of the first radiation source employed to create excitations in the electron system, given pumping radiation of the first radiation source with 1.537 eV, at power levels of 20, 100, and 300 μW (from bottom to top). X peak indicates the transition from 0LL of heavy holes to the higher spin sublevel of the 0LL of electrons, and X' is associated with the creation of excitation in the electron system. **b** Amplitude of photoinduced resonant reflection measured for the fixed photon energies of 1.537 and 1.5332 eV of the first and second radiation sources, respectively, at a constant power level of the second source E₂ and varied power level of the first source. The linewidth of the photoinduced resonant reflection line is defined by the slit of the spectrometer used to register the scattered radiation. **c** Dependence of the photoinduced resonance reflection intensity (taken from **b**) on the power level of the first radiation source.

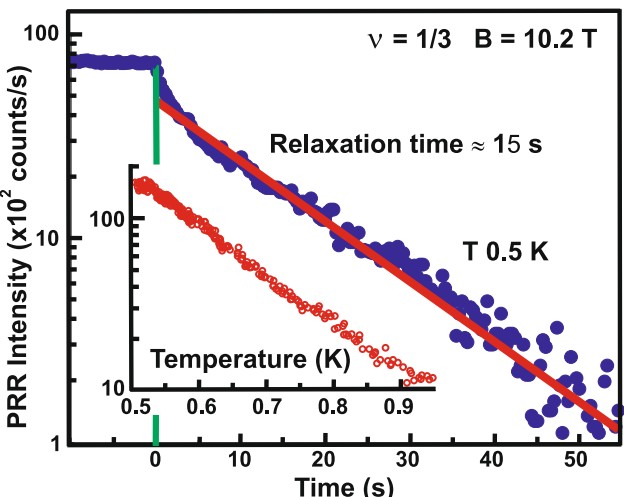

**Fig. 6 Relaxation time for the neutral excitations.** Dependence of the amplitude of photoinduced resonant reflection signal (at 1.5332 eV energy of the scattered photons) on the time duration after the turn-off of the first radiation source (with 1.537 eV photon energy of pumping radiation) employed to create excitations in the electron system (blue dots). Green line indicates the moment of shutting down the exciting beam. Red line is a linear approximation on a logarithmic scale. The inset shows a photoinduced resonant reflection intensity as a function of temperature, given the fixed power levels of both radiation sources. The signal of photoinduced resonant reflection vanishes approximately at the same temperature as when the Laughlin state $\nu = 1/3$ is destroyed.

the lowest one for all momenta (Supplementary Fig. 4). In this case, the combination of both spin one and spin-zero magneto-rotons with net-zero momentum (spin one "magneto-graviton") will have lower energy than the spin-zero "magneto-graviton." In other words, given zero momentum, spin exciton with single-particle Zeeman energy and spin one "magneto-graviton" will be the lowest-energy neutral excitations.

**Experiment.** The sample of size of approximately $3 \times 3$ mm² was placed into a pumped cryostat with liquid ³He, which in turn was placed into a ⁴He cryostat with a superconducting solenoid. The setup allowed measurements at the bath temperature down to 0.45 K and at the magnetic field up to 14 T. Optical measurements were made using the dual-fiber technique with the use of multimode quartz glass optical fiber with a core diameter of 400 μm and numerical aperture N.A. of 0.39. One fiber was used for photoexcitation, and a second for collecting the emission from the sample and transferring it onto the entrance slit of a grating spectrometer equipped with a CCD camera cooled by liquid nitrogen.

Considering practical resources available to the authors of this paper, in creating an electron system with quasi-equilibrium spin excitations, the optimal magnetic field for the Laughlin state $\nu = 1/3$ is close to 10 T. Hence, the characteristic width of a quantum well with a two-dimensional electron system is chosen to be about 20 nm or less (Supplementary Fig. 4). The advantage of using quantum wells of the width under 20 nm is that their valence band turns out to be less complex than that of wider wells. The former case is related to more significant energy splitting of light- and heavy-hole subbands. Also, in the narrower wells, the splitting of the size quantized subbands is larger for the heavy and light holes alike. Although these peculiarities do not affect the Laughlin state $\nu = 1/3$ directly, they become of critical importance in the analysis of allowed optical transitions from the conduction band to the valence band. A detailed description of these aspects is provided in the Supplementary Materials.

For the reasons mentioned above, to form the quasi-equilibrium ensemble of neutral excitations with spin one in the Laughlin state $\nu = 1/3$, we chose the quantum well of 18-nm width, with the electron density of $8.4 \cdot 10^{10}$ cm⁻². The magnetic field of the Laughlin state $\nu = 1/3$ is about 10.2 T. The electron mobility of the system is $3.5 \cdot 10^6$ cm²/V s, which is sufficient for observing the quantization of Hall's conductivity $\nu = 1/3$ at the temperature of 0.5 K. The test sample was placed into a cryostat with pumped ³He vapor, ensuring the lowest sample temperature of 0.45 K and the magnetic field of up to 14 T (Fig. 1). The system was excited with a pair of tunable cw (continuous wave) laser sources, where the radiation of one of the sources could be modulated by an external mechanical shutter. The experiments were carried out using photoluminescence and photoinduced resonant reflection optical techniques.

Two continuous-wave tunable lasers with narrow spectral widths of emission lines (20 and 5 MHz) were employed as optical sources, which enabled us to use one of the lasers for resonant excitation of the electron system and the other for recording spectra of resonance reflectance and photoluminescence. The experimental geometry was chosen when measuring the resonance reflectance spectra so that the specularly reflected beam axis coincided with the receiving fiber axis at an angle of incidence of ~10°. The contribution of the sample surface reflection was suppressed using crossed linear polarizers set between the sample and the ends of the pumping and collecting fibers. One of the lasers was used as an optical pump of the system via the excitation of the electrons. The pumping laser intensity was limited to a power below 0.1 mW to minimize heating effects. An emission from a probing laser, which was weaker by an order of magnitude, was coupled into the same waveguide. The resonance reflectance spectrum was obtained by scanning the emission wavelength of the probing laser and registering the laser line intensity using the spectrometer with the CCD camera. The photoinduced resonant reflection was obtained as the difference in the resonance reflectance spectra with the resonance pump switched on and off.

Photoluminescence was employed to achieve two different goals. First, taking advantage of the excitation process used in photoluminescence, we formed the ensemble of neutral excitations as follows. Upon the excitation of an electronic system, a photon of certain energy is absorbed in the quantum well. As a result, an electron from the valence band passes to the upper spin sublevel of zero Landau level in the conduction band (Fig. 1a). Due to strong spin-orbit coupling in GaAs valence band, the photoexcited hole rapidly relaxes to the lowest spin sublevel of the heavy-hole subband in a time of about 100 ps (the relaxation times of photoexcited holes for the system under study were measured independently using time-resolved photoluminescence). Since only the lowest spin sublevel is occupied at the equilibrium, the recombination of a photoexcited hole with an equilibrium electron in the lowest spin sublevel of zero Landau level in conduction band leads to the formation of a spin one neutral excitation in the electronic system, which consists of a photoexcited electron at the upper spin Landau sublevel and a hole at the lower one. In all optical processes, the longitudinal momentum of a photon was chosen equal to zero, making possible two types of spin one neutral excitations—a spin exciton and a spin "magneto-graviton" (Fig. 1a).

The second objective of using photoluminescence was to assess the change in the optical signal under the conditions of cw excitation due to the appearance of neutral excitation and to select among possible neutral excitations those which could cause such a change, based on the preliminary information about their properties. It is known, for example, that under cw excitation of an electron system with a reasonable radiation power density (not leading to overheating of the entire system), it is impossible to accumulate enough spin excitons to produce an experimentally detectable change in the photoluminescence signal. The lifetime of spin excitons is just too short as they relax to the ground state in times on the order of 100 ns or less[28]. However, to form a macroscopic quasi-equilibrium ensemble of neutral excitations with densities at the level of 1–10 percent of the number of equilibrium electrons (the signal that can be detected against the background photoluminescence from the filled-in equilibrium lowest spin sublevel of the zero Landau level), the characteristic lifetime of excitations should reach the values of 10 μs or more[21,32].

## Data availability
The processed numerical data files are available at the public GitHub repository (https://github.com/lilymusina/NatCommun2021). The plots on data generated in this study by the numerical model are provided in the Supplementary Information. All other data that support the plots within this paper and other findings of this study are available from the corresponding author upon reasonable request.

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

## Acknowledgements
We thank Boris Kaisin for assistance in experiment, S.M. Dickmann and V.D. Kulakovsky for fruitful discussion. The work was supported by the Russian Science Foundation, Grant 18-12-00246.

## Author contributions
L.V.K., A.S.J., E.I.B. and A.A.Z. performed measurements, L.I.M., A.B.V. and O.V.V. performed numerical simulation. V.Y.U. conducted the molecular beam epitaxy growth. L.V.K., A.S.Z., L.I.M., A.B.V. and I.V.K. developed the concept of experiment and analyzed the obtained data. L.V.K. and L.I.M. wrote the manuscript.

## Competing interests
The authors declare no competing interests.
