## [Peer Review File · Nature Communications]

REVIEWER COMMENTS

Reviewer #1 (Remarks to the Author):

Laughlin anyon complexes with Bose properties

Even after three decades since the discovery of fractional quantum Hall states, keep unwinding itself to reveal its complex many-body physics and thus intrigues condensed matter physicist till today. Magneto-rotons, the charge-neutral excitations in 2-dimensional electron gas have similarities to the very fascinating quantum excitations in gravitational field a.k.a. gravitons. The inelastic light scattering by Pinczuk (ref 12 of the paper) sees the resonance with zero momentum in photoluminescence spectra may be the first signature of gravitons in condensed matter physics. In this work, the authors have performed an experiment (additional support from the computational model) in the line of Pinczuk et al. photoluminescence experiment on $\nu = 1/3$. Their extended work experimental results agree with previous works and claimed two big footings from their experimental results:

1. Observation of quantum statistics of the magneto-gravitons complexes i.e. they exhibit bosonic quantum statistics.
2. The direct measurement of the relaxation time of the neutral excitation setting record of ~ 10 seconds, longest relaxation time for quantum excitations observed in the (fractional) quantum Hall states.

I enjoyed the reading of the research article. The data presented in the research article are convincing and well-bounded but still, I have some reservations and comments, especially regarding the main claim: Bosonic statistics for the magneto-gravitons neutral excitations. The statement is not bounded by experimental demonstration. How author could claim that the ensemble of states in the quantum well structure will be a single quantum state and excitation observe in the experiment is due to the occupation of a single quantum state? –As far as my understating, the resonance in the photoluminescence (PL) experiment does not require the quantum degeneracy but merely a coherence property of an ensemble state. What would be the role of disorder and localized states on their PL resonances, especially ones closer to the interfacial edges of the quantum well?

Additionally, I have some minor comments:

1. What is photon excitation energy for the PL data shown in Fig 2? Also, why for the $\nu = 0.44$ ($\sim \nu = 3/7$?) and $\nu = 1/3$ data should be explained in better way. It is misleading at first glance why $\nu = 0.44$ though a very fragile state still shows the peak in PL spectra at a high temperature which is not the case.
2. The statement in the text “the photoluminescence intensity from the upper spin sublevel is an order of magnitude higher than that from the lowest spin sub level filled with electrons in equilibrium”, could be explained better with high-temperature data and could be also compared with $\nu = 1$ data.

3. Can authors provide similar data for $\nu = 1$ state which is like $\nu = 1/3$ state instead it will have electronic states compared to composite fermionic state for the latter?
4. Please specify the experimental parameters in the figure captions. For example, in Fig 4, the power and wavelength of the probe are missing.
5. Figure 5 (c) which is stacked between Fig 5 (a) and Fig 5 (b), y scale is misleading. Could authors, explain the splitting of the peak (X and X') in Fig 5 with the increase in power of the source excitations.
6. In Figure 6, why the value of amplitudes in the inset and main figure is different?
7. What is CDE in Fig 2 of the extended data? Also, the inset cartoon figure showing magneto-graviton and magneto-roton are misleading. Please write the abbreviation (for example MR for magneto-roton etc.) under the figure or show it with an arrow to its feature in the figure.
8. The authors also claim that the spin one magneto-roton has the lowest neutral excitation energy and well-founded by the simulation. Could the author present the experimental data for the wider quantum well (QW > 20 nm) to show how PL resonance differs in this case? Maybe a statement to compare the current results with one observed by Pinczuk (ref 12) will be encouraging.
9. Lastly, the paper includes lots of terms related to the magneto-rotors for example, spin one magneto-roton, spin-zero magneto-roton, spin-one magneto-gravitons, etc. It becomes very difficult to follow up especially text becomes very confusing (see methods: computation). Maybe authors should consider explaining these terms in the supplementary information with descriptive cartoons. Simplification of the text is thus very much required.

With this, I will say that the experimental results are interesting but still arguments are not supported well by the experimental results. If authors could support this with the additional experimental results, I would be happy to review the work again.

Reviewer #2 (Remarks to the Author):

The authors use numerical methods and optical techniques to study neutral excitations associated with the Laughlin state at $\nu=1/3$. First, they use the numerical scheme to argue that the spin one excitations are the lowest energy excitation branch. Then, they show that a feature in the photoluminescence spectrum emerges at low temperature at $\nu=1/3$ ($T < 1.6\text{K}$), which appears insensitive to the excitation energy. Based on this observation, the authors draw a conclusion that a condensate of neutral excitation is formed. In addition, the amplitude of the photoinduced resonant reflection line grows stronger with an increasing power level of the radiation source, which is argued to be evidence for the Bose statistics. Lastly, they show that the relaxation time of the neutral excitation is longer than 10 seconds, which is the most intriguing result.

Although the experimental data presented here is interesting, the manuscript overall is somewhat disorganized and difficult to read. The narrative jumps back and forth, sometimes citing results that have not been discussed as support for making a statement. In addition, the general disorganization makes it unclear which conclusion is made based on what observation. As such, some of the key statements of this manuscript appear incoherent. As such, a substantial re-write is needed before the manuscript is publishable.

Here, I will first detail my major concerns.

1. what is the experimental evidence for spin one neutral excitation? It appears that this claim is based on numerical results. If this is the case, it should be clarified in the manuscript. While numerical results offer helpful insights when compared with experimental data, they do not offer definitive proof alone. Therefore, the authors should clearly specify whether a conclusion is drawn based on the numerical results or experimental observations, instead of treating numerical and experimental results on equal footings.

2. related to point 1, discussion regarding the neutral excitation in the method section is confusing:

“We find that, it is possible for the spin zero magneto-roton to be the lowest energy excitation in the region of roton minimum. On the other hand, there is also a possibility that the energy of the spin one excitation branch can be the lowest one for all momenta (Extended Data Fig. 4). In this case, the combination of both spin one and spin zero magneto-rotors with net zero momentum (spin one “magneto-graviton”) will have lower energy than the spin zero “magneto-graviton.” In other words, given zero momentum, spin exciton with single-particle Zeeman energy and spin one “magneto-graviton” will be the lowest-energy neutral excitations”

This is the first main result presented in the manuscript, it deserves more discussion in the main text. Also, I do not follow the logic behind this argument.

3. The authors cited the lifetime for spin excitation as evidence for spin one magneto gravitons without further clarification (line 105-107). If this is the main result, it should be further discussed in the main text. In addition, the discussion in the method section does not provide clarification. What are relevant measurements of this spin excitation lifetime and what is the signature of spin one magneto graviton?

4. what is the evidence for a condensate of neutral excitations? The authors made a statement on line 102-103:

“based on measured photoluminescence spectra, we can draw a qualitative conclusion that under cw photoexcitation, there is formed a condensate of neutral excitations, exhibiting collective response to a photoexcited hole in the valence band.”

It is unclear what constitutes a “qualitative conclusion”, but the assertion of a condensate phase appears inconclusive. The authors will need to elaborate more to make such a claim. In addition, a condensate implies Bose statistics, which is the conclusion of fig5 (and maybe fig6). However, bose

statistics do not guarantee a condensate phase at 0.5K. It would be helpful if the authors can present all experimental evidence and discuss their implication, before drawing conclusions.

Several minor points:

Figures, especially schematics are confusing. For example, schematics in Figure 1 are not explained, apart from giving each panel a name. It will be helpful to add more explanation in the caption or main text, elaborate on their relevance to the manuscript, and maybe also provide references.

Reviewer #3 (Remarks to the Author):

The manuscript "Laughlin anyon complexes with Bose properties" is devoted to the development a numerical scheme of solving Schrodinger's equation for the electron system with a finite number of particles in line with previous studies and studying the dispersion properties of neutral excitations for the Laughlin state $\nu = 1/3$ in GaAs/AlGaAs quantum wells. Given the selected electron system with necessary parameters, by means of optical techniques (photoluminescence and photoinduced resonant reflection), the authors succeeded in forming a macroscopic ensemble of neutral spin one excitations in the Laughlin state $\nu = 1/3$ and developed a method for the direct measurement of their relaxation time. According to the reported results, the ensemble of spin one neutral excitations exhibits the properties of a Bose system.

In summary, the results of the calculations presented in this manuscript are new and exciting. The theoretical approach used for calculations corresponds to the modern level of the theory. Finally, I recommend accepting this manuscript for publication in Nature Communications.

Response to referees

On behalf of all the authors of the manuscript “Laughlin anyon complexes with boson properties” I genuinely thank all the reviewers for providing us with their valuable comments, thus allowing us to make the manuscript more comprehensible for the reader and more rigorous in terms of the facts that we are presenting. It was a great joy to find out about the interest towards our work from our colleagues, and we hope this has been only the start of a fruitful discussion in the scientific community.

Comments from Reviewer #1

Comment 1: I enjoyed the reading of the research article. The data presented in the research article are convincing and well-bounded but still, I have some reservations and comments, especially regarding the main claim: Bosonic statistics for the magneto-gravitons neutral excitations. The statement is not bounded by experimental demonstration. How author could claim that the ensemble of states in the quantum well structure will be a single quantum state and excitation observe in the experiment is due to the occupation of a single quantum state?

As far as my understating, the resonance in the photoluminescence (PL) experiment does not require the quantum degeneracy but merely a coherence property of an ensemble state. What would be the role of disorder and localized states on their PL resonances, especially ones closer to the interfacial edges of the quantum well?

Response: The referee has pointed out a crucial moment in our work. We claim bosonic **properties of the anyonic complexes**. By the bosonic property we mean the obvious experimental fact that the more magneto-gravitons are excited, the easier it is to excite an extra magneto-graviton in the same quantum state. The role of disorder and interfacial edge modes has not been a subject of the present study, but can be an interesting next step in the development of this research.

Comments from Reviewer #2

Comment 1: what is the experimental evidence for spin one neutral excitation? It appears that this claim is based on numerical results. If this is the case, it should be clarified in the manuscript. While numerical results offer helpful insights when compared with experimental data, they do not offer definitive proof alone. Therefore, the authors should clearly specify whether a conclusion is drawn based on the numerical results or experimental observations, instead of treating numerical and experimental results on equal footings.

Response: The referee has drawn attention to an important fact of the research. We have studied carefully possible optical transitions in our system (see Supplementary). From polarization measurements and corresponding simulations we have found that only optical transitions at the upper spin sublevel of the zero Landau level of the conductance band are detected in the experiment. That is meant we could excite excitations with spin 1 only. The clarification of this fact and link to supplementary is added to the main text.

Comment 2: related to point 1, discussion regarding the neutral excitation in the method section is confusing:

“We find that, it is possible for the spin zero magneto-roton to be the lowest energy excitation in the region of roton minimum. On the other hand, there is also a possibility that the energy of the spin one excitation branch can be the lowest one for all momenta (Extended Data Fig. 4). In this case, the combination of both spin one and spin zero magneto-rotions with net zero momentum (spin one “magneto-graviton”) will have lower energy than the spin zero “magneto-graviton.” In other words, given zero momentum, spin exciton with single-particle Zeeman energy and spin one “magneto-graviton” will be the lowest-energy neutral excitations”

This is the first main result presented in the manuscript, it deserves more discussion in the main text. Also, I do not follow the logic behind this argument.

Response: There are no logical conclusions presumed, we do not claim receiving an analytical description or construction of a theory. All our calculations are numerical, not analytical. We only present experimental and numerical data and look forward to future collaboration with theorists on this issue.

Comment 3: *The authors cited the lifetime for spin excitation as evidence for spin one magneto gravitons without further clarification (line 105-107). If this is the main result, it should be further discussed in the main text. In addition, the discussion in the method section does not provide clarification. What are relevant measurements of this spin excitation lifetime and what is the signature of spin one magneto graviton?*

Response: The lifetime of a spin exciton was measured in ref.31. The spin exciton lifetime is of the order of 100ns. The theory concerning it can also be found in A. S. Zhuravlev, S. Dickmann, L. V. Kulik, and I. V. Kukushkin, Slow spin relaxation in a quantum Hall ferromagnet state, Phys. Rev. B 89, 161301(R)

The citation of ref. 31 was added on line 105-107

Comment 4: *what is the evidence for a condensate of neutral excitations? The authors made a statement on line 102-103:*

“based on measured photoluminescence spectra, we can draw a qualitative conclusion that under cw photoexcitation, there is formed a condensate of neutral excitations, exhibiting collective response to a photoexcited hole in the valence band.”

It is unclear what constitutes a “qualitative conclusion”, but the assertion of a condensate phase appears inconclusive. The authors will need to elaborate more to make such a claim. In addition, a condensate implies Bose statistics, which is the conclusion of fig5 (and maybe fig6). However, bose statistics do not guarantee a condensate phase at 0.5K. It would be helpful if the authors can present all experimental evidence and discuss their implication, before drawing conclusions.

Response: We completely agree with the reviewer. There is no proper evidence of condensation, therefore we have changed “a condensate of neutral excitations” to “dense ensemble of neutral excitations exhibiting collective response”.

Comments from Reviewer #3

The manuscript “Laughlin anyon complexes with Bose properties” is devoted to the development a numerical scheme of solving Schrodinger’s equation for the electron system with a finite number of particles in line with previous studies and studying the dispersion properties of neutral excitations for the Laughlin state $\nu = 1/3$ in GaAs/AlGaAs quantum wells. Given the selected electron system with necessary parameters, by means of optical techniques (photoluminescence and photoinduced resonant reflection), the authors succeeded in forming a macroscopic ensemble of neutral spin one excitations in the Laughlin state $\nu = 1/3$ and developed a method for the direct measurement of their relaxation time. According to the reported results, the ensemble of spin one neutral excitations exhibits the properties of a Bose system.

In summary, the results of the calculations presented in this manuscript are new and exciting. The theoretical approach used for calculations corresponds to the modern level of the theory. Finally, I recommend accepting this manuscript for publication in Nature Communications.

Response: We highly appreciate the high assessment of our work. We plan on providing more research using this method and its developments, and it is a great pleasure to see this approach acknowledged.

List of changes in manuscript

The changes are marked yellow in the text.

1. (Reviewer #1 minor comment 1)
What is photon excitation energy for the PL data shown in Fig 2?
The photon excitation energy is added in Fig.2 caption.
2. (Reviewer #1 minor comment 4)
Please specify the experimental parameters in the figure captions. For example, in Fig 4, the power and wavelength of the probe are missing.
The changes have been made.
3. (Reviewer #1 minor comment 5)
Figure 5 (c) which is stacked between Fig 5 (a) and Fig 5 (b), y scale is misleading. Could authors, explain the splitting of the peak (X and X') in Fig 5 with the increase in power of the source excitations.
The figure has been changed, and the description of peaks X and X' is added into caption.
4. (Reviewer #1 minor comment 7)
What is CDE in Fig 2 of the extended data? Also, the inset cartoon figure showing magneto-graviton and magneto-roton are misleading. Please write the abbreviation (for example MR for magneto-roton etc.) under the figure or show it with an arrow to its feature in the figure.
The abbreviations have been added for Extended Data Fig.2 and Extended Data Fig.3.
5. (Reviewer #1 minor comment 9)
the paper includes lots of terms related to the magneto-rotors for example, spin one magneto-roton, spin-zero magneto-roton, spin-one magneto-gravitons, etc. It becomes very difficult to follow up especially text becomes very confusing (see methods: computation). Maybe authors should consider explaining these terms in the supplementary information with descriptive cartoons.
More details on classification of excitations are now given in Methods. The cartoons are now present on Extended Data Fig.2 and Extended Data Fig.3.
6. (Reviewer #2 comment 1)
The more detailed explanation and link to supplementary is added to the main text (line 114-115).
7. (Reviewer #2 comment 3)
The citation of ref. 31 was added on line 105-107
8. (Reviewer #2 comment 4)
There is no proper evidence of condensation, therefore in the line 102-103 and all the other places we have changed “a condensate of neutral excitations” to “dense ensemble of neutral excitations exhibiting collective response”.
9. (Reviewer #2 minor comments)
schematics in Figure 1 are not explained, apart from giving each panel a name. It will be helpful to add more explanation in the caption or main text, elaborate on their relevance to the manuscript, and maybe also provide references.
The changes have been made in Fig.1 caption and Methods section.

Additional clarifications

1. (Reviewer #1 minor comment 1)

What is photon excitation energy for the PL data shown in Fig 2? Also, why for the $\nu = 0.44$ ($\sim \nu = 3/7$?) and $\nu = 1/3$ data should be explained in better way. It is misleading at first glance why $\nu = 0.44$ though a very fragile state still shows the peak in PL spectra at a high temperature which is not the case.

We present data for $\nu = 0.44$ in comparison with $\nu = 1/3$ to highlight that we can distinguish the $\nu = 1/3$ spectra from another filling factor by observing the peak that is destroyed by high temperature (when the $1/3$ state is destroyed). For $\nu = 0.44$ we see no difference between high and low temperature.

2. (Reviewer #1 minor comment 2, Reviewer #1 minor comment 3)

The statement in the text “the photoluminescence intensity from the upper spin sublevel is an order of magnitude higher than that from the lowest spin sub level filled with electrons in equilibrium”, could be explained better with high-temperature data and could be also compared with $\nu = 1$ data.

The high temperature data in comparison with the low temperature are now in Fig.3. The thorough discussion of the filling factor $\nu = 1$ is out of the scope of the present research. It has been done by the authors in a distinct publication (ref. 31)

3. (Reviewer #1 minor comment 6)

In Figure 6, why the value of amplitudes in the inset and main figure is different?

These are two different figures with the same units on vertical scale. We pointed on that in the caption.

4. (Reviewer #1 minor comment 8)

The authors also claim that the spin one magneto-roton has the lowest neutral excitation energy and well-founded by the simulation. Could the author present the experimental data for the wider quantum well ($QW > 20$ nm) to show how PL resonance differs in this case? Maybe a statement to compare the current results with one observed by Pinczuk (ref 12) will be encouraging.

We have performed the similar measurements for the wide wells (350nm) and found out the absence of long-living excitations. However, we hesitate about the closure of this question, as the lowest available temperature we can obtain is not low enough to make safe conclusions about the result for such a wide well (The coulomb interaction is much weaker in this case). Considering Pinczuk's paper (ref.12), the data presented there are indeed for well width 25nm, however the method is inelastic light scattering is quite different from PRR, so the direct comparison of them is hardly possible.

5. The text was revised as suggested by reviewers, but since English is not a native language of the authors, we do hope that an additional editing is available at Nature Communications and we are ready to pay for this kind of assistance.

We appreciate your time and effort, and look forward to your response.

REVIEWERS' COMMENTS

Reviewer #1 (Remarks to the Author):

I found the reply by the authors satisfactory, and I do recommend the paper for publication.

On a side note, I found few typos in the paper, but I presume these will be corrected in final proofreading. For example, please see the text below:

"We form a macroscopic quasi-equilibrium ensemble of neutral excitations - spin one anyon complexes in the Laughlin state $\nu = 1/3$, experimentally, where is the electron filling factor."

where is the electron filling factor -> where ν is the filling factor(?)

"Besides, all experimentally observed optical transitions from the zero Landau level of heavy holes of the valence band occur"

-> Besides, all experimentally observed optical transitions from the zero Landau level of heavy 'holes' of the valence band occur

Reviewer #2 (Remarks to the Author):

The authors have adequately addressed my comments. The updated manuscript is easy to read, the reported results are new and exciting. Based on these considerations, I recommend publication in Nature Communication.

Reviewer #3 (Remarks to the Author):

This revised version of manuscript contains the responses to referee's questions and suggestions. I am satisfied by these revisions in the manuscript.

Finally, I recommend accepting this revised version of manuscript for publication in Nature Communications.